# Measurement of violence against women and disability: protocol for a scoping review

Sarah R. Meyer  ,[1] Molly E. Lasater,[2] Lindsay Lee,[3] Claudia Garcia-Moreno[1]

[1]Department of Sexual and Reproductive Health and Research, WHO, Geneva, Switzerland
[2]Department of Mental Health, Johns Hopkins University Bloomberg School of Public Health, Baltimore, Maryland, USA
[3]Department of Noncommunicable Diseases, WHO, Geneva, Switzerland

**Correspondence to**
Dr Claudia Garcia-Moreno;
garciamorenoc@who.int

## ABSTRACT

**Introduction** Violence against women is a serious threat to women's health and human rights globally. Disability has been associated with increased risk of exposure to different forms of violence, however, there are questions concerning how best to measure this association. Research on understanding the association between violence and disability among women has included incorporating short disability measures into violence against women prevalence surveys. The potential to improve understanding of interconnections between violence and disability by measuring violence within disability-focused research is underexplored. The scoping review described here focuses on three areas of measurement of violence against women and disability: (1) measurement of violence within the context of disability-focused research, (2) measurement in research focused on the intersection of disability and violence and (3) measurement of disability in the context of research focused on violence against women. Specifically, we aim to map definitions, measures and methodologies in these areas, globally.

**Methods and analysis** For our scoping review, we will conduct searches for quantitative studies of disability-focused research which use measures of violence against women, and measures of disability in research focused on violence against women, in 11 online databases. Two authors will independently review titles and abstracts retrieved through the search strategy. We will search for grey literature, search the websites of National Statistics Offices for all countries to identify any national or subnational disability research and consult with experts for input. Data extraction will be conducted independently by one author and reviewed by another author, and data will be analysed and synthesised using a thematic synthesis approach.

**Ethics and dissemination** Ethics approval was not sought as no primary data is being collected. Findings will be disseminated through a publication in a peer-reviewed journal, through coordinated dissemination to researchers, practitioners, data users and generators and through various working groups and networks on violence against women and disability.

### Strengths and limitations of this study

⇒ This scoping review is designed with a comprehensive search strategy, including a structured search strategy for country-level and regional data that are unpublished in peer-reviewed literature;
⇒ This scoping review focuses on a significant gap in the evidence, and provides an approach to mapping and understanding available measurement methods of violence against women used in studies of disability and measurement methods of disability used in studies of violence against women;
⇒ This scoping review uses an appropriate search strategy, data extraction and analysis to comprehensively map the global field of measurement of violence against women and disability.

long-lasting physical and mental health impacts of violence against women constitute a global public health threat. An expanding evidence-base has identified a number of risk factors for women's exposure to violence, such as economic factors, including poverty, patterns of asset ownership and wealth inequalities,[1] social norms concerning male authority over female behaviour and norms justifying violence against women[2] and exposure to childhood abuse and exposure to intimate partner violence against one's mother as a child.[3]

A potential risk factor that is currently poorly understood is disability. In particular, while it is hypothesised that disability may increase women's vulnerability to violence (and violence can also lead to disability), there is limited evidence concerning the intersection between disability and violence against women. A systematic review and meta-analysis of prevalence and risk of violence against adults with disabilities found that adults with disabilities are at increased risk of violence compared with adults without disability. However, the review did not conduct sex-stratified analyses to identify if gender dimensions compounds the risk of violence against persons with disabilities.[4]

## BACKGROUND

Violence against women is a serious threat to women's health and human rights globally. The wide range of severe and often

Disability may be a risk factor for exposure to violence against women for a range of reasons. Studies have suggested that violence against women with disabilities is greater than violence against women without disabilities due to perpetrator-related characteristics. For example, women with disabilities are more likely to have partners who hold views supporting patriarchal dominance, and to be possessive and jealous, leading to enacting violence within the context of intimate partner relationships.[5] Qualitative studies have identified specific vulnerabilities to violence experienced by women with disabilities, including reliance on partners for support in daily activities and/or for financial support, lack of social support and lack of availability of accessible services for violence prevention and response for women with disabilities.[6] However, the evidence-base concerning disability as a risk factor for women's experience of violence is relatively limited. Target 5.2 of the Sustainable Development Goals (SDGs) is, "End all violence against and exploitation of women and girls", and the overarching goal of the SDGs is to "leave no-one behind". Within the framework of the SDGs, there is a need for a strengthened evidence-base concerning violence against women and disability, to inform violence prevention and policy response to violence against women and ensure effective design and implementation of policies, services and programmes.[7] Developing and strengthening this evidence-base requires rigorous study design and measurement, and this scoping review emerges from the need to improve understanding of appropriate and effective measures and methodologies to shed light on the intersection between disability and violence against women.

Some studies and reviews have addressed the question of disability as a risk factor for women's experiences of violence. For example, the *What Works to Prevent Violence against Women and Girls Global Programme* included the Washington Group Short Set of Questions on Disability (Washington Group Questions) in quantitative impact assessments. Analyses from baseline assessments for interventions in six countries (Afghanistan, Bangladesh, Ghana, Nepal, South Africa and Tajikistan) showed that women with disabilities are between two to four times more likely to experience intimate partner violence than women without disabilities.[8] A systematic review of studies exploring the intersection of violence and intellectual disabilities identified five qualitative and one mixed-methods study, concluding that the evidence is extremely sparse, and that 'the current state of knowledge concerning the use and experience of partner violence by adults with intellectual disabilities is fundamentally inadequate, and until this knowledge gap is closed, our ability to provide appropriate evidence-based services to both perpetrators and victims is limited'.[9] A literature review focused on prevalence of interpersonal violence against persons with disabilities found the data indicated consistent associations of disability with a higher exposure to lifetime and past 5-year intimate partner violence among women.[10] Women's experience of violence was not a

specific focus in all of these reviews. A literature review of qualitative and quantitative studies addressing the question of prevalence and risk factors for violence among women with acquired disabilities, identified specific risk factors such as physical, economical and emotional dependency, and explored the role of social isolation in vulnerability to violence among women with disabilities.[11]

A significant challenge in understanding disability as a risk factor for women's experience of violence is how to quantify this association, both in terms of measures of disability and of violence. Disability can also be a consequence of intimate partner or other forms of violence against women, and the relationship between violence and disability may be bidirectional. For example, pre-existing disabilities may be a risk factor for violence victimisation, and women's experiences of violence may lead to disability, which entails that the relationship between violence and disability can be difficult to disentangle. Further compounding this challenge are the issues with disability assessment tools. Different conceptual models of disability are linked to different measurement approaches.[12] Studies of disability globally employ vastly different definitions of disability or cut-offs to determine disability status across studies, impacting prevalence estimates and comparability of data sources.[13] There are some measures, such as Washington Group Questions, that have specifically been developed to address issues of comparability. They reflect an approach that assesses functioning, and whether persons with disability are able to participate in society.[14] However, research has indicated that the Washington Group Questions do not reliably identify individuals who screen positive clinically for moderate or greater impairment.[15] Use of the Washington Questions for screening has been found to define individuals with mild-to-moderate clinical impairments as non-disabled.[16]

Population-based prevalence studies of violence against women are a primary way of generating data on prevalence of, risk factors for and health outcomes of violence against women. Recent donor interest in and support of strengthening evidence concerning violence against women with disabilities has focused on incorporating disability questions within population-based national violence against women prevalence surveys, for example, in Timor-Leste and Mongolia.[17 18] However, there are methodological and practical challenges to using violence-focused prevalence studies to understand how disability is associated with violence. Cross-sectional prevalence studies do not enable assessment of whether self-reported disability is a risk factor for greater exposure to violence, or whether increased exposure to violence leads to higher (self-reported) disability among women. Sampling strategies in violence against women prevalence studies are household-based, and therefore exclude women with disabilities who may be living in other settings (ie, institutions, group housing). Women with profound and severe disabilities are usually excluded from violence against women prevalence surveys, and there are challenging ethics concerns

regarding interviewing women with specific disabilities that may impair communication or cognition.[19] In addition, some evidence indicates that women with disabilities may be exposed to different forms of violence and perpetrators than are traditionally captured in violence-focused research.[8 20 21] Therefore, measurement instruments presently used in violence-focused research may not adequately capture the range of types and perpetrators of violence against women with disabilities.

## AIM

This scoping review seeks to strengthen and support efforts to understand the linkages and intersections between disability and violence against women, specifically by mapping definitions, measures and methodologies in quantitative literature on this topic.

Violence against women is defined by the United Nations as "any act of gender-based violence that results in, or is likely to result in, physical, sexual or mental harm or suffering to women, including threats of such acts, coercion or arbitrary deprivation of liberty, whether occurring in public or in private life".[22] The WHO defines disability as described in the International Classification of Functioning, Disability and Health (ICF): disability is the outcome of the interaction between (1) one's health conditions and (2) contextual factors such as physical accessibility of the environment, access to assistive products or attitudes of others. To describe a person's disability status under the ICF framework, it is necessary to examine both components of this definition.[23]

Our scoping review will map definitions, measures and methodologies in three areas of measurement of violence against women and disability: (1) measurement of violence within the context of disability-focused research, (2) measurement in research focused on the intersection of disability and violence and (3) measurement of disability in the context of research focused on violence against women. We focus on quantitative literature given our scoping review emerges from data requirements for the SDGs and seeks to address current developments in quantitative population-based surveys of violence against women. For the purpose of our review, we define disability-focused research as quantitative research seeking to estimate the prevalence of disability or identify associations between disability and other health outcomes. We define research focused on the intersection of disability and violence as research that focuses on associations between disability and violence, without being solely focused on either disability or violence as an outcome. This focus on three distinct, but overlapping, areas of literature is designed to inform current debates and discussions regarding how to generate evidence concerning violence against women with disabilities. As noted above, for example, donors' interest in understanding the association between disability and violence against women has focused on incorporating measures of disability within national violence against women surveys. However, a broader characterisation of which measures of disability and violence are used and available, how definitions are operationalised and what methodologies are feasible and appropriate is needed.

## METHODS

Arksey and O'Malley lay out a framework for methods of scoping reviews that we draw on in design of our protocol.[24]

We will conduct a scoping review of studies published in peer-reviewed literature, and grey literature, including studies conducted or published by national statistical offices, WHO and other United Nations agencies. A scoping review is the most appropriate review method for studies that have exploratory research questions.[25] Scoping reviews, described as commonly used for 'reconnaissance', are specifically useful in contexts "where a body of literature has not yet been comprehensively reviewed, or exhibits a large, complex or heterogeneous nature," as is the case in the body of literature in question.[26] Scoping reviews are typically useful for mapping a field "in terms of its nature, features and volume", and given the state of knowledge and existing available evidence syntheses, this is the most appropriate type of review to address our research questions. In contrast to a systematic review, which focuses on a specific question, or set of questions, with a more tightly limited field of enquiry, a scoping review takes a broader approach to focus on mapping the literature and clarifying key concepts, enabling greater breadth than a systematic review. For this review, we seek to map the field of measurement of violence against women and disability in different bodies of literature, identify measures used, research gaps and explore feasibility of developing research objectives for a systematic review.[24]

The following sections on search strategy, data searches, inclusion and exclusion criteria and selection of studies correspond to Arksey and O'Malley's Stage 2, Identifying relevant studies, and Stage 3, Selection of studies.

### Search strategy

We identified the following domains as part of the research question: disability; women; violence; and quantitative research. For each of these domains, we identified the relevant keywords and search terms, which vary by database (Ttable 1). The search strategy will be appropriately modified for each database, including syntax and specific terms, topics and/or headings. The search has not been limited by year of publication or type of publication. An expert librarian at the WHO provided advice on search strategy and selection of databases.

### Data sources

Data sources for the searches included following electronic databases: PubMed, PsycINFO, Embase, CINAHL, Web of Science, PILOTS, Sociological Abstracts, ERIC, AgeLine, Social Work Abstracts, International Bibliography of the

| Table 1 | PubMed search strategy |
|---|---|
| 1 | "Intellectual disability"(MeSH) OR "Disabled persons"(MeSH) OR "Communication disorders"(MeSH) OR "Developmental disabilities"(MeSH) OR "Mentally Disabled persons"(MeSH) OR "physical disability"(MeSH) OR "physically disabled"(TIAB) OR "intellectual disabilit*"(TIAB) OR "handicap"(TIAB) OR "functional impairment"(TIAB) OR "mental disorder*"(TIAB) OR "mentally disabled"(TIAB) OR "mental disability*"(TIAB) |
| 2 | Women(MeSH) OR female(MeSH) OR wife(TIAB) OR spouses(MeSH) OR wives(TIAB) OR "female partner*"(TIAB) OR spouse*(TIAB) |
| 3 | "Elder abuse"(MeSH) OR "domestic violence"(MeSH) OR "Intimate Partner Violence"(MeSH) OR "violence"(MeSH) OR "battered women"(MeSH) OR "violence"(MeSH) OR "aggression"(MeSH) OR "spouse abuse"(MeSH) OR "Physical Abuse"(MeSH) OR Rape (MeSH) OR "elder neglect"(TIAB) OR "elder mistreatment"(TIAB) OR "elder maltreatment"(TIAB) OR "assault"(TIAB) OR "sexual assault"(TIAB) OR "rape"(TIAB) OR "sexual abuse"(TIAB) OR "psychological abuse"(TIAB) OR "psychological violence"(TIAB) OR "emotional abuse"(TIAB) OR "emotional violence"(TIAB) OR "neglect"(TIAB) OR "economic abuse"(TIAB) OR "verbal abuse"(TIAB) OR "violence against women"(TIAB) OR "abused women"(tiab) OR "intimate terrorism"(tiab) OR "marital rape"(tiab) OR "wife beating"(tiab) OR "relationship aggression"(tiab) |
| 4 | "epidemiologic methods"(MeSH) OR "Comparative Study"(Publication Type) OR "outcome and process assessment (health care)"(Mesh) OR "statistics and numerical data"(Subheading) OR "Evaluation Studies"(Publication Type) OR "meta analysis"(Publication Type) OR "multicenter study"(Publication Type) OR "incidence"(TIAB) OR "surveillance"(TIAB) OR "prevalence"(TIAB) OR "epidemiology"(subheading) OR "Health Care Evaluation Mechanisms"(Mesh) OR "morbidity"(TIAB) OR "burden"(TW) OR "Cross sectional study"(MeSH) OR "case-control studies"(MeSH) OR "Cohort studies"(MeSH) OR "Surveys and questionnaires"(MeSH) OR "cross-sectional stud*"(TIAB) OR "quantitative survey"(TIAB) OR "survey"(TIAB) |
| 1 AND 2 AND 3 AND 4 | |

Social Sciences, Social Services Abstracts, ProQuest Criminal Justice, ASSIA, Dissertations & Theses Full Text, and Dissertations & Theses Global. The grey literature search will be conducted by one author (SRM), who will conduct structured google searches: 'Country X disability survey', 'Country X disability study' and 'Country X disability statistics', for each country, reviewing 10 pages of results per search.

We will search the websites of National Statistics Offices for all countries to identify any national or subnational disability research, as well as national violence against women studies and Demographic and Health Surveys that have included both disability and violence against women modules. We will also review data and reports on disability available to the WHO, which includes data from the WHO Model Disability Survey. We will identify experts in the field of research on violence and/ or disability measurement, including researchers, practitioners and policymakers, and contact them to provide any relevant literature. All experts will be contacted at least two times to provide the research team with additional resources to consider for inclusion. We will review the reference list of existing relevant systematic reviews and scoping reviews to identify relevant publications.

### Inclusion and exclusion criteria

Studies will be eligible for this scoping review if the study:
i. Uses a quantitative methodology; mixed methods studies will be included if the quantitative data are reported separately;
ii. Compares women with disability to women without disability (studies including men and women with disability will be included if sex-specific analyses are included) OR includes only women with disability;
iii. Assesses exposure to any form of violence;
iv. Examines violence experienced as an adult, aged 15 and older (studies including violence experienced before the age of 15 will be included if violence experienced above 15 is also measured).

Fifteen and above is selected as the age cut-off as this is lower age-limit for relevant SDG indicators regarding elimination of violence against women and girls. Non-English language articles will be included depending on number and capacity of team, which includes members who are fluent in Spanish, French and Portuguese.

Studies will not be eligible for this scoping review if the study only:
i. Focuses only on common mental health disorders (depression, anxiety, post-traumatic stress disorder (PTSD));
ii. Compares women with disability to men with disability;
iii. Only focuses on violence experienced before the age of 15;
iv. Uses data from case studies or client files;
v. Is based on caregiver report and/or forensic exam;
vi. Focuses on validity/reliability of the measure or scale development.

These exclusion criteria were developed to ensure that the identified literature addresses the specific study aims and identify a body of literature that allows for contrast and comparison to answer the key research questions.

Mental disorders are often considered a part of disability. However, specifically in the area of violence against women, there is a robust evidence-base concerning the associations between common mental disorders (depression, anxiety and PTSD) and violence against women including several systematic reviews and meta-analyses.[27–29] Given the aim of this scoping review to focus on an area of measurement and methodology that is far less well-developed, we are limiting the breadth of our scoping review by excluding studies focusing solely on common mental disorders.

We will identify characteristics of studies (published and grey literature) meeting inclusion criteria, with a focus on mapping and evaluating measures of violence used in this research, identifying types of violence assessed, instruments used and specifics of measures (ie, perpetrator, time frame). This review differs from previous reviews of violence against persons with disability by focusing on: (1) women, (2) any setting (community, institution, for example), (3) any type of violence and perpetrator, (4) measurement of disability (which measures, how measured) and (5) measurement of violence (which measures, how measured).

### Data management
EndNote V.X9 will be used as our bibliographic software management platform. We will remove duplicates using EndNote, prior to exporting titles and abstracts to an Excel spreadsheet for review. Data extraction results will be recorded in separate Excel spreadsheets. A flow diagram will be presented in any final publications, showing results of each stage of the review and adhering to the PRISMA (Preferred Reporting Items for Systematic Reviews and Meta-Analyses) statement.

### Selection of studies
Two authors will independently review titles and abstracts retrieved through the search strategy, to determine which should be included for full-text review. If an abstract or title is considered relevant by either of the authors, it will be included for full-text review. Two authors will independently review all articles selected for full-text review for eligibility, to reach consensus on inclusion in the review. Any discrepancies will be resolved with the input of the third team member. Reasons for excluding articles will be recorded.

### Data extraction
After full-text review, the following data will be extracted from all included articles using a standardised data extraction form: country studied; research questions; study design (comparing individuals with disability vs people without disability); sampling method and sample characteristics; data collection (measurement method);

disability measurement (definitions, measurement (self-report, instrument), measure of severity, functional impairment); violence measurement (definition, types measured, perpetrators, time frame, instruments use); data analysis methods; risk and protective factors; main findings (as reported by the study's own authors); ethical considerations and discussion of disability and violence specific issues; and any reported study limitations. The data extraction tool was designed specifically for this scoping review, and as such, includes necessary variables to address the aims of the study.

Data extraction will be conducted independently by one author (SRM), and accuracy of the data extraction checked by a second author, with discrepancies resolved by discussion and, if necessary, by discussion with another author (CG-M) to reach consensus. We will develop and pre-test a data extraction spreadsheet, to be used to compile a summary of characteristics and key findings of the included studies. The spreadsheet will also include categories relevant to data synthesis, described further below. We will not conduct quality assessment, given this is a scoping review. Pham *et al* (2014) note that one of the distinctions between a systematic review and scoping review is that a scoping review aims to describe available material without critical appraisal of studies, and therefore, quality assessment is less necessary and common in scoping reviews.[30] This data extraction process corresponds with Stage 4, Charting the data, in Arksey and O'Malley's framework.

### Data synthesis
We will present results of the search and data extraction, using both simple quantitative summaries (ie, tabulation of % of studies from each region, % of studies that used specific sampling methods), and narrative synthesis of the studies, which includes highlighting similarities and differences in the measures of disability and of violence employed in the included studies, and exploration of other patterns in aspects of study design and measurement methodologies in included studies.[31] This phase corresponds with Arksey and O'Malley's Stage 5, Collating, summarising and reporting the results. Figure 1 displays all components of the study process.

### Patient and public involvement
Patients were not involved in the development of this scoping review. Members of the public were not consulted specifically for the development of the research questions, however, previous research and consultations with experts has indicated that this is a relevant and important area of enquiry in the field of violence against women research.

### DISCUSSION
This manuscript describes a protocol for a scoping review of global measurement of violence against women within the context of disability-focused research and vice versa.

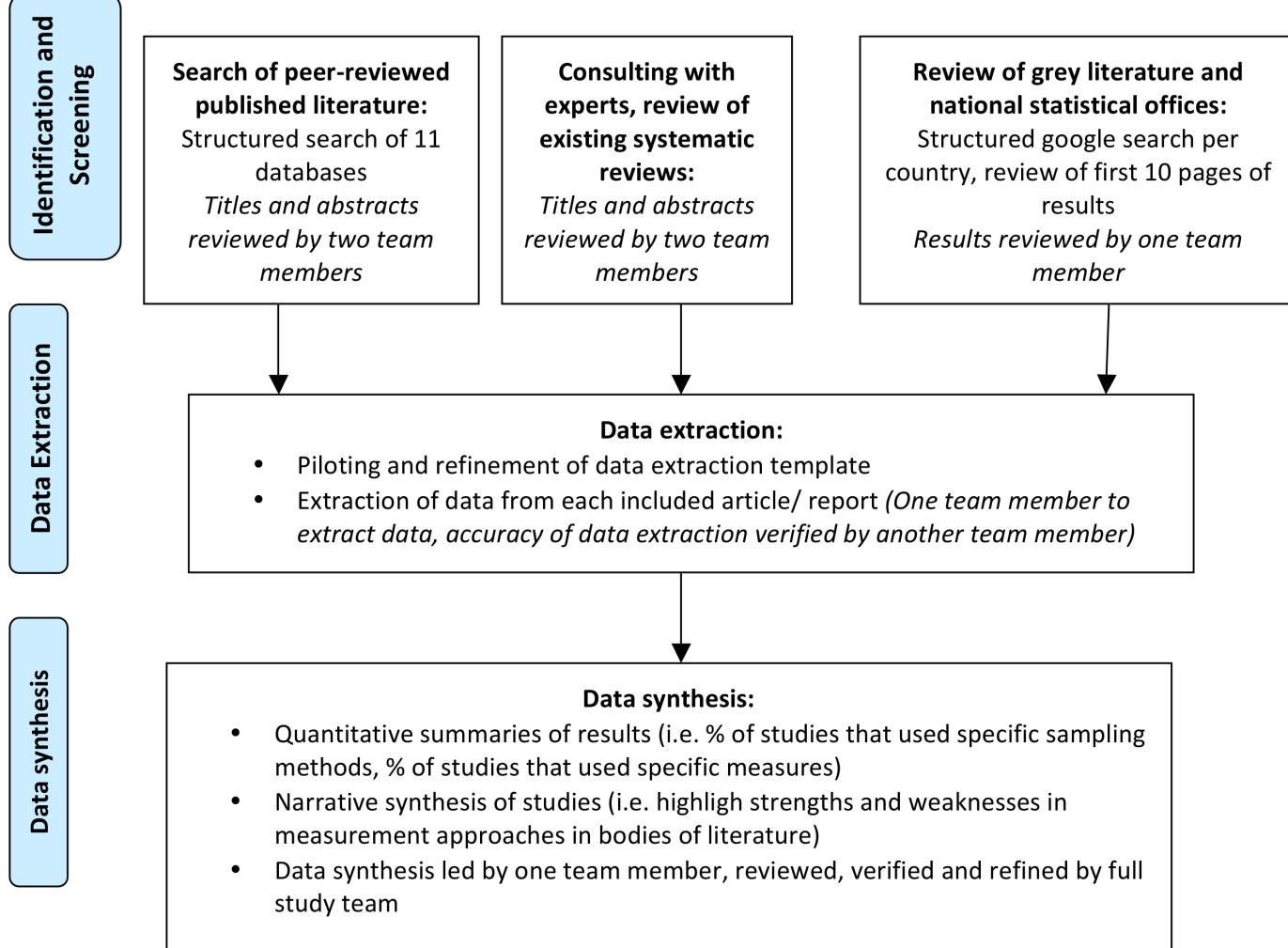

**Figure 1** Study process.

Strengths of the review include a rigorous and expansive search strategy, including disability and violence against women studies conducted by National Statistical Offices and not published in peer-reviewed literature; a clear and structured process of data extraction; and a focus on generating a map of available measures and methodologies assessing the intersection of violence and disability, within a body of evidence that has not been rigorously reviewed. The scoping review is designed to assess global literature, explicitly using search methods to ensure that studies conducted in low- and middle-income countries are included. This will contribute to the discussion on improving the ways of measuring the intersections of disability and violence against women.

Improved understanding and expanded evidence on how disability and violence against women intersect is needed to develop effective evidence-based programming and policy for prevention and response to violence against women globally. Various aspects of lived experiences of disability may influence types of violence experienced, access to and utilisation of services and ways in which research, policy and programming on violence against women can be adapted or refined to adequately

address the needs of women with disability who experience violence. Yet, as a report from a recent expert consultation on measurement of violence against women with disabilities noted, 'far more work needs to be done to establish appropriate, effective, agreed and internationally comparable methods for measuring many of the structural, institutional and interpersonal forms of violence that women with disabilities experience on a daily basis'.[19] Expanding understanding of linkages and intersections between disability and violence against women requires further consideration of how disability and violence are currently being assessed, including what type(s) of measures are being used and within what type(s) of methodologies and study designs.

A primary limitation of this review is the definition of disability that is utilized . The WHO recognises that disability includes "impairments, activity limitations, and participation restrictions," and that disability "is a complex phenomenon, reflecting the interaction between features of a person's body and features of the society in which he or she lives" (https://www.who.int/topics/disabilities/en/). According to this definition, knowledge of the environment in which a person lives is critical

to understanding someone's experience of disability. However, to keep the scope of the review manageable and with the understanding that research on the environmental component of disability is often lacking, we will use search terms for disability that primarily focus on impairments or specific health conditions that are known to cause particular impairments. This may limit the literature identified and included in the search and bias results towards research focused on only the bodily, health or impairment component of disability. A limitation of the process of study screening and selection is that one team member will conduct the grey literature search and identification process. This may limit the rigour of the grey literature search process, but will also enable review of more grey literature for possible inclusion (ie, 10 pages of results for three separate searches). A limitation of the study design is the focus on quantitative literature. Qualitative descriptions are important components of understanding women's experiences of disability and violence. However, the focus of this scoping review is specifically on quantitative measurement. This is motivated by the existing state of the field of evidence and need for data to answer key questions to inform prevention and response in violence against women policy and programmes. A separate review of qualitative literature could complement the current study.

## Ethics and dissemination

Findings of the review will be used to inform recommendations regarding evidence-generation on disability and violence against women. Findings will be with shared researchers, practitioners, data users and generators with an interest in violence against women. We will also share results with members of the Technical Advisory Group to the Interagency Working Group on Violence against Women Estimation and Data, a group of experts on measurement of violence against women and global violence against women data convened by the Department of Sexual and Reproductive Health and Research, WHO. Final outcomes will be presented in a manuscript for publication in a peer-reviewed journal. This will be disseminated through the Interagency Working Group above and other partners. The Sensory Functions, Disability and Rehabilitation Team will also disseminate through their networks focused on disability, including through the Interagency Support Group for the Convention on the Rights of Persons with Disabilities. Dissemination will also engage with disability advocacy groups, through the International Disability Alliance.

**Contributors** SRM and CG-M designed and developed the scoping review, which was conceptualised by CG-M. SRM, MEL and LL developed and refined search strategies with input from CG-M. SRM prepared the manuscript with substantive input from all other authors. All authors reviewed the manuscript prior to submission.

**Funding** The scoping review will be conducted with funding from the Department for International Development for the UN Women-WHO Joint Programme on Strengthening Methodologies and Measurement and building national capacities for violence against women data through the UNDP-UNFPA-UNICEF-WHO-World Bank Special Programme of Research, Development and Research Training in Human Reproduction (HRP), a cosponsored programme executed by the WHO.

**Competing interests** None declared.

**Patient and public involvement** Patients and/or the public were not involved in the design, or conduct, or reporting, or dissemination plans of this research.

**Patient consent for publication** Not required.

**Provenance and peer review** Not commissioned; externally peer reviewed.

**ORCID iD**
Sarah R. Meyer http://orcid.org/0000-0001-8595-2358

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
