## [Reviewer comments · BMJ Open]

ARTICLE DETAILS

TITLE (PROVISIONAL)	Measurement of violence against women and disability: protocol for a scoping review
AUTHORS	Meyer, Sarah; Lasater, Molly; Lee, Lindsay; Garcia-Moreno, Claudia

VERSION 1 – REVIEW

REVIEWER	Jennifer MacGregor Western University, Canada
REVIEW RETURNED	16-Jun-2020

GENERAL COMMENTS	This manuscripts clearly describes a sound plan for conducting a scoping review on an important topic. I have no major concerns – a few finicky things are described below. 1. (page 10) I found the following sentences unclear – it talks about people with moderate to severe impairment not being reliably identified in the first sentence but then in the second, it also seems that people with mild to moderate impairment are not identified properly. “However, research has indicated that the Washington Group questions do not reliably identify individuals who screen positive clinically for moderate or severe impairment 13. Use of the Washington Questions for screening has been found to define individuals with mild to moderate disability as non-disabled.”2. (p. 12) Although I am familiar with scoping reviews and find the first Methods paragraph clear, some readers may not be familiar with scoping reviews – perhaps because the term scoping review is sometimes used for reviews that should be labeled otherwise. I suggest adding (either in text or footnote) a definition of a scoping review (e.g., Peters et al. 2015 uses the word ‘reconnaissance’ to describe it, which I find helpful).3. (p. 12) Typo: CINALH should be CINAHL.4. (p. 13) Suggest there “will be no data limits applied” rather than there “are no date limits.”5. (p.15) I think it’s fine that there will be no quality assessment of included articles, but readers not familiar with scoping reviews may wonder about it. I suggest providing at least one reference to support this decision, e.g.: Pham, M. T., Rajic, A., Greig, J. D., Sargeant, J. M., Papadopoulos, A., & McEwen, S. A. (2014). A scoping review of scoping reviews: Advancing the approach and
---

	enhancing the consistency. Research Synthesis Methods, 5, 371–385.  6. (p. 15) On this page it specifies that data extraction will be done by one person and verified by another, but in the abstract it only says it will be done by one person. If you have enough room to add the info regarding verification to the abstract as well, that would be ideal so that it does not seem like your plan is unclear. 7. (p. 16) Suggest “Members of the public” instead of “Public”. There are a few places like this throughout where it seemed the authors were writing a bit ‘short’ (similar to how you might drop words and write more concisely for an abstract). I suggest checking for and fixing these throughout. 8. (p. 17) First sentence under ‘Ethics and dissemination’ – I think the word ‘shared’ is missing. This sentence is also really long (can the explanation of the advisory group perhaps be in a footnote?).
--	--

REVIEWER	Rebecca J. Macy University of North Carolina at Chapel Hill, United States
REVIEW RETURNED	26-Jun-2020

GENERAL COMMENTS	This manuscript describes the protocol for a scoping review, which is focused on the measurement of violence within the context of disability-focused research, research focused on the intersection of disability and violence, and measurement of disability in the context of research focused on violence against women. As the authors rightfully state in the manuscript, the issues of disability in the context of violence against women are pressing topics that have been woefully under-investigated to-date. Accordingly, a state of the summary of the current evidence may make an important contribution. In addition, there has, so far, been limited development and publications describing study protocols in the context of violence against women research. For all these reasons, this manuscript has promising potential to make a notable contribution. In addition, there are also ways that the manuscript might be strengthened, which I detail below. INTRODUCTION & AIMS  • Please consider clearly operationalizing the key phrases/words “violence against women” and “disability” early in the manuscript. • Related to the point above, the study’s rationale for focusing on the intersecting issues of women, disability, and violence is not entirely clear as the manuscript now reads. Please consider clarifying the rationale, especially given that violence can take many forms, disabilities are diverse and varied, and that men with disabilities as well as people with nonbinary gender identities with disabilities are also at increased risk for violence. Please know that with this point, I am not arguing that the current focus is a limitation, rather that the study’s rationale is not yet fully clear. • Overall, the manuscript’s introduction touches on many significant gaps in the literature, but not always in ways that have a logical flow that helps to highlight for readers the most critical gaps in current global research. • The authors’ point that the relationship between disability and violence can be bidirectional is important and worth elaboration (page 9).
--

	 • Please clarify the meaning of the phrase “model of disability” (page 9). • Please explicate the relationship among the three stated aims of the review. In addition, the authors could fruitfully clarify and provide a research-based rationale for why these three aims are especially pressing and significant. Also, in places, the aims seem to contradict one another. Specifically, is the study most focused on investigating violence within the context of disability-focused research or disability in the context of research focused on violence against women or both? • The scoping review seems to be taking a global focus, which is in my view a study strength that could be fruitfully made explicit. • Please explain the study’s rationale for a scoping review rather than a systematic or another review approach. • Please explicate the rationale for the study’s focus on quantitative studies. METHODS  • Please consider defining what is meant by “other international offices.” • Please consider providing detail about the non-English capabilities and capacities of the team. • Please provide a clear rationale and explanation for the database selection. • Please provide a rationale for the age of 15 as a cut-off in the inclusion criteria. • To echo a comment from above, explain what is meant by “disability,” including why mental illnesses are being excluded from this review. The rationale for this study decision is not yet clear to this reader. Please know that with this point, I am not arguing that the current focus is a limitation. Rather, I am requesting clarity. • Some of the exclusion criteria do not seem to follow from the study’s aims: i.e., Utilizes data from case studies or client files; Is based on caregiver report and/ or forensic exam; and Focuses on validity/ reliability of the measure or scale development. • Please describe the research process for identifying and locating the grey literature. In particular, how will rigor and comprehensiveness be assured? • Please consider explicating how the data extraction form will ensure that the necessary data will be collected so that the study’s aims can be met. • Please consider whether a diagram or figure of the study protocol might help readers better understand the study. DISCUSSION  • Overall, the manuscript’s discussion touches on many significant issues, but not always in ways that have a logical flow that helps to highlight for readers the most important contributions this study will make to the global research. • Please consider whether the study’s exclusive focus on quantitative studies might be a limitation. • Given the disability community’s call to do “nothing about us without us,” the team’s limited plans to engage with disability advocacy groups, as well as with people with disabilities themselves in the study process and its dissemination seems like a potential missed opportunity or study limitation.
--	---

REVIEWER	Nicole Doria Dalhousie University, Canada
REVIEW RETURNED	03-Jul-2020

GENERAL COMMENTS	Research Question/Study Objective: The aims for the study are vast. Exploring the relationships between disability and violence against women is different than mapping definitions, measures and methodologies used to measure violence against women, in the context of disability-focused research. I am not sure which is the focus of the scoping review. Would clarify with what the research question is that you are looking to answer. Methods: Include who developed the search strategy/if a librarian or expert was consulted to do so. Good use of data sources. What is the justification for 15 and older? A 15 year olds experience will likely be quite different than a 30 year olds experience. For data management. Covidence or a similar program would be helpful to complete the screening process. I suggest following the steps of Arksey and O'Malley for methods and outlining your methods with the use of their steps to conducting a scoping review, or another framework for conducting a scoping review. Include a section for study limitations.
---

VERSION 1 – AUTHOR RESPONSE

Reviewer 1:

1. (page 10) I found the following sentences unclear – it talks about people with moderate to severe impairment not being reliably identified in the first sentence but then in the second, it also seems that people with mild to moderate impairment are not identified properly. “However, research has indicated that the Washington Group questions do not reliably identify individuals who screen positive clinically for moderate or severe impairment 13. Use of the Washington Questions for screening has been found to define individuals with mild to moderate disability as non-disabled.”

We have edited this for clarity and it now reads:

“However, research has indicated that the Washington Group questions do not reliably identify individuals who screen positive clinically for moderate or greater impairment. Use of the Washington Questions for screening has been found to define individuals with moderate or greater clinical impairments as non-disabled.”

2. (p. 12) Although I am familiar with scoping reviews and find the first Methods paragraph clear, some readers may not be familiar with scoping reviews – perhaps because the term scoping review is sometimes used for reviews that should be labeled otherwise. I suggest adding (either in text or footnote) a definition of a scoping review (e.g., Peters et al. 2015 uses the word ‘reconnaissance’ to describe it, which I find helpful).

We agree that a definition of a scoping review would be useful, and have added the following definition within the Methods section:

“A scoping review is the most appropriate review method for studies that have exploratory research questions²³. Scoping reviews, described as commonly used for “reconnaissance,” are specifically useful in contexts “where a body of literature has not yet been comprehensively reviewed, or exhibits a large, complex or heterogeneous nature,” as is the case in the body of literature in question in this scoping review²⁴. Scoping reviews are typically useful for mapping a field “in terms of its nature, features and volume,” and given the state of knowledge and existing available evidence syntheses, this is the most appropriate type of review to address our research questions. In contrast to a systematic review, which focuses on a specific question, or set of questions, with a more tightly limited field of enquiry, a scoping review takes a broader approach to focus on mapping the literature and clarifying key concepts, enabling greater breadth than a systematic review.”

3. Typo: CINALH should be CINAHL

Corrected.

4. (p. 13) Suggest there “will be no data limits applied” rather than there “are no date limits.”

This phrase refers to date limits set on the search, i.e. only from 1990 onwards. We see that we have included this earlier, and so have removed this sentence.

5. (p.15) I think it's fine that there will be no quality assessment of included articles, but readers not familiar with scoping reviews may wonder about it. I suggest providing at least one reference to support this decision, e.g.: Pham, M. T., Rajic, A., Greig, J. D., Sargeant, J. M., Papadopoulos, A., & McEwen, S. A. (2014). A scoping review of scoping reviews: Advancing the approach and enhancing the consistency. *Research Synthesis Methods*, 5, 371–385.

We have added the following sentence: Pham et al. (2014) note that one of the distinctions between a systematic review and scoping review is that a scoping review aims to describe available material without critical appraisal of studies, and therefore, quality assessment is less necessary and common in scoping reviews²⁶

6. (p. 15) On this page it specifies that data extraction will be done by one person and verified by another, but in the abstract it only says it will be done by one person. If you have enough room to add the info regarding verification to the abstract as well, that would be ideal so that it does not seem like your plan is unclear.

We have added this to the abstract, and the sentence now reads: “Data extraction will be conducted independently by one author and reviewed by another author, and data will be analysed and synthesised using a thematic synthesis approach.”

7. Suggest “Members of the public” instead of “Public”. There are a few places like this throughout where it seemed the authors were writing a bit ‘short’ (similar to how you might drop words and write more concisely for an abstract). I suggest checking for and fixing these throughout.

We have changed the phrase to read “members of the public,” and reviewed the full text to expand in some places for clarity.

8. (p. 17) First sentence under ‘Ethics and dissemination’ – I think the word ‘shared’ is missing. This sentence is also really long (can the explanation of the advisory group perhaps be in a footnote?).

Corrected and edited for clarity.

Reviewer 2:

1. Please consider clearly operationalizing the key phrases/words “violence against women” and “disability” early in the manuscript.

We agree that this would be useful, and have added definitions of disability and violence against women in the first section Aims, which now reads:

“Violence against women is defined by the United Nations as “any act of gender-based violence that results in, or is likely to result in, physical, sexual or mental harm or suffering to women, including threats of such acts, coercion or arbitrary deprivation of liberty, whether occurring in public or in private life”²¹. The World Health Organization defines disability as described in the International Classification of Functioning, Disability and Health [ICF]: disability is the outcome of the interaction between (1) one’s health conditions and (2) contextual factors such as physical accessibility of the environment, access to assistive products, or attitudes of others. To describe a person’s disability status under the ICF framework, it is necessary to examine both components of this definition²².”

2. Related to the point above, the study’s rationale for focusing on the intersecting issues of women, disability, and violence is not entirely clear as the manuscript now reads. Please consider clarifying the rationale, especially given that violence can take many forms, disabilities are diverse and varied, and that men with disabilities as well as people with nonbinary gender identities with disabilities are also at increased risk for violence.

We agree that there are a wide range of forms of violence and disability, and that violence is perpetrated against women, men and non-binary individuals. The focus on women in this scoping review emerges from the specific imperatives to address this global challenge of public health and inform more effective prevention and response mechanisms.

We have clarified the rationale in the following ways:

- Included discussion of the impetus of the SDGs – both in terms of elimination of violence against women and ‘leaving no one behind’; the scoping review is embedded within the need for evidence spurred by the SDGs;

- Donor interest in understanding the intersection of disability and violence against women has largely focused on integrating measurement of disability within violence against women surveys; it is not clear that this is appropriate or effective, and this scoping review seeks to identify other measures and methodologies that shed light on this intersection;

The following added texts in the Introduction addresses these aspects of the rationale:

“Target 5.2 of the Sustainable Development Goals is, “End all violence against and exploitation of women and girls,” and the over-arching goal of the SDGs is to “leave no-one behind.” Within the framework of the SDGs, there is a need for a strengthened evidence-base concerning violence against women and disability, to inform violence prevention and policy response to violence against women and ensure effective design and implementation of policies, services and programs ⁷. Developing and strengthening this evidence-base requires rigorous study design and measurement, and this scoping review emerges from the need to improve understanding of appropriate and effective measures and methodologies to shed light on the intersection between disability and violence against women.”

“Recent donor interest in and support of strengthening evidence concerning violence against women with disabilities has focused on incorporating disability questions within population-based national violence against women prevalence surveys, for example, in Timor Leste and Mongolia ^{17 18}.”

3. Overall, the manuscript’s introduction touches on many significant gaps in the literature, but not always in ways that have a logical flow that helps to highlight for readers the most critical gaps in current global research.

We have reviewed and restructured the Introduction, which now is now structured around the following main points:

- Disability may be a risk factor for women’s experience of violence, but there is limited evidence and we need more understanding of this to improve policy and prevention;
 - Existing data and reviews on disability as a risk factor for violence – what we do know;
 - The challenge of measuring this association, especially in terms of available measures of disability; and
 - The limitations of assessing disability in the context of violence against women prevalence surveys.
4. The authors’ point that the relationship between disability and violence can be bidirectional is important and worth elaboration (page 9).

We have elaborated on this and included the following:

“For example, pre-existing disabilities may be a risk factor for violence victimization, and women’s experiences of violence may lead to disability, which entails that the relationship between violence and disability can be difficult to disentangle. Further compounding this challenge are the with disability assessment tools.”

5. Please clarify the meaning of the phrase “model of disability” (page 9).

We have added the word “conceptual” to clarify.

6. Please explicate the relationship among the three stated aims of the review. In addition, the authors could fruitfully clarify and provide a research-based rationale for why these three aims are especially pressing and significant. Also, in places, the aims seem to contradict one another. Specifically, is the study most focused on investigating violence within the context of disability-focused research or disability in the context of research focused on violence against women or both?

We have reviewed and restructured the Introduction, Study Aims and Methods to better reflect that the study is focused on both violence in the context of disability research, and disability in the context of violence research.

For example, the Aims section now reads:

“This scoping review seeks to strengthen and support efforts to understand the linkages and intersections between disability and violence against women, specifically by mapping definitions, measures and methodologies in quantitative literature on this topic.

Violence against women is defined by the United Nations as “any act of gender-based violence that results in, or is likely to result in, physical, sexual or mental harm or suffering to women, including threats of such acts, coercion or arbitrary deprivation of liberty, whether occurring in public or in private life”²². The World Health Organization defines disability as described in the International Classification of Functioning, Disability and Health [ICF]: disability is the outcome of the interaction between (1) one’s health conditions and (2) contextual factors such as physical accessibility of the environment, access to assistive products, or attitudes of others. To describe a person’s disability status under the ICF framework, it is necessary to examine both components of this definition²³.

Our scoping review will map definitions, measures and methodologies in three areas of measurement of violence against women and disability: i) measurement of violence within the context of disability-focused research, ii) measurement in research focused on the intersection of disability and violence, and iii) measurement of disability in the context of research focused on violence against women. We focus on quantitative literature given our scoping review emerges from data requirements for the SDGs and seeks to address current developments in quantitative population-based surveys of violence against women. For the purpose of our review, we define disability-focused research as quantitative research seeking to estimate the prevalence of disability or identify associations between disability and other health outcomes. We define research focused on the intersection of disability and violence as research that focuses on associations between disability and violence, without being solely focused on either disability or violence as an outcome. This focus on three distinct, but overlapping, areas of literature is designed to inform current debates and discussions regarding how to generate evidence concerning violence against women with disabilities. As noted above, for example, donors’ interest in understanding the association between disability and violence against women has focused on incorporating measures of disability within national violence against women surveys. However, a broader characterization of which measures of disability and violence are used

and available, how definitions are operationalized, and what methodologies are feasible and appropriate is needed.”

7. The scoping review seems to be taking a global focus, which is in my view a study strength that could be fruitfully made explicit.

We have included the following sentence in the Discussion:

“The scoping review is designed to assess global literature, explicitly using search methods to ensure that studies conducted in low and middle-income countries are included.”

8. Please explain the study’s rationale for a scoping review rather than a systematic or another review approach.

We have added further description of a scoping review in the Methods section, which describes why a scoping review is the most appropriate type of review for these research questions:

“Scoping reviews, described as commonly used for “reconnaissance,” are specifically useful in contexts “where a body of literature has not yet been comprehensively reviewed, or exhibits a large, complex or heterogeneous nature,” as is the case in the body of literature in question in this scoping review ²⁴. Scoping reviews are typically useful for mapping a field “in terms of its nature, features and volume,” and given the state of knowledge and existing available evidence syntheses, this is the most appropriate type of review to address our research questions. In contrast to a systematic review, which focuses on a specific question, or set of questions, with a more tightly limited field of enquiry, a scoping review takes a broader approach to focus on mapping the literature and clarifying key concepts, enabling greater breadth than a systematic review. For this review, we seek to map the field of measurement of violence against women and disability in different bodies of literature, identify measures used, research gaps and explore feasibility of developing research objectives for a systematic review ²⁵.”

9. Please explicate the rationale for the study’s focus on quantitative studies.

We have added this sentence:

“We focus on quantitative literature given our scoping review emerges from data requirements for the SDGs and seeks to address current developments in quantitative population-based surveys of violence against women.”

10. Please consider defining what is meant by “other international offices.”

We have changed this to read, “other United Nations agencies.”

11. Please consider providing detail about the non-English capabilities and capacities of the team.

We have added this: “ Non-English language articles will be included depending on number and capacity of team, which includes members who are fluent in Spanish, French and Portuguese.”

12. Please provide a clear rationale and explanation for the database selection.

We were advised on database selection by a librarian at the WHO. We have added the following sentence: “An expert librarian at the World Health Organization provided advice on search strategy and selection of databases.”

13. Please provide a rationale for the age of 15 as a cut-off in the inclusion criteria.

We have added the following rationale:

“15 and above is selected as the age cut-off as this is lower age-limit for relevant SDG indicators.”

14. To echo a comment from above, explain what is meant by “disability,” including why mental illnesses are being excluded from this review. The rationale for this study decision is not yet clear to this reader.

We have clarified that we are excluding *common* mental disorders (depression, anxiety and post-traumatic stress disorder). The body of evidence underpinning the association between these disorders and violence against women is very robust, and issues associated with measures of these disorders are substantially different than other forms of disability that are included in this review. As such, we have clarified why these disorders are excluded from this scoping review, and cited some of the evidence-base concerning this association.

We have added the following text:

“Mental disorders are often considered a part of disability. However, specifically in the area of violence against women, there is a robust evidence-base concerning the associations between common mental disorders (depression, anxiety and PTSD) and violence against women including several systematic reviews and meta analyses^{27 28 29}. Given the aim of this scoping review to focus on the an area of measurement and methodology that is far less well-developed, we are limiting the breadth of our scoping review by excluding studies focusing solely on common mental disorders.”

15. Some of the exclusion criteria do not seem to follow from the study’s aims: i.e., Utilizes data from case studies or client files; Is based on caregiver report and/ or forensic exam; and Focuses on validity/ reliability of the measure or scale development.

These exclusion criteria were developed to ensure that we identify the relevant body of literature. Without these specific exclusion criteria, we are concerned that the review will generate a body of literature that contains sub-sets of studies that are non-comparable.

Specifically:

- **Case studies or client files** have different aims and methodologies than survey literature, making comparison of measures, methodologies, etc difficult to assess;
- **Caregiver report** in the field of violence measurement is considered flawed/ unreliable in many ways; while we want to include a broad range of violence measures, the gold standard is self-report and we do not wish to include studies that rely on caregiver report;
- **Forensic exam for assessment of violence is a separate field of measurement**; within the context of our study rationale (to inform evidence-generation to support SDGs, prevention and response to violence against women), especially in low-resources contexts, forensic exam methodologies;

- **Validity and reliability of measures or scale development** – design of validity and reliability studies is fundamentally designed to address a different research question. This body of literature could fruitfully be addressed in a separate review.

We have included the following sentence to broadly characterize this:

“These exclusion criteria were developed to ensure that the identified literature addresses the specific study aims and identify a body of literature that allows for contrast and comparison to answer the key research questions.”

16. Please describe the research process for identifying and locating the grey literature. In particular, how will rigor and comprehensiveness be assured?

We have added the following description:

“The grey literature search will be conducted by one author (SM), who will conduct structured google searches: “Country X disability survey,” “Country X disability study” and “Country X disability statistics,” for each country, reviewing 10 pages of results per search.”

We have added the following in the Discussion section regarding the limitation of this search and selection process being conducted by one author:

“A limitation of the process of study screening and selection is that one team member will conduct the grey literature search and identification process. This may limit the rigor of the grey literature search process, but will also enable review of more grey literature for possible inclusion (i.e. 10 pages of results for 3 separate searches).”

17. Please consider explicating how the data extraction form will ensure that the necessary data will be collected so that the study’s aims can be met.

We have added the following sentence:

“The data extraction tool was designed specifically for this scoping review, and as such, includes necessary variables to address the aims of the study.”

18. Please consider whether a diagram or figure of the study protocol might help readers better understand the study.

We have included a Figure to display the study process.

19. Overall, the manuscript’s discussion touches on many significant issues, but not always in ways that have a logical flow that helps to highlight for readers the most important contributions this study will make to the global research.

We have restructured the Discussion section to have study limitations at the end, for more logical flow.

20. Please consider whether the study’s exclusive focus on quantitative studies might be a limitation.

We have added the following to the Discussion section:

“A limitation of the study design is the focus on quantitative literature. Qualitative descriptions are important components of understanding women’s experiences of disability and violence. However, the focus of this scoping review is specifically on quantitative measurement. This is motivated by the existing state of the field of evidence and need for data to answer key questions to inform prevention and response in violence against women policy and programs. A separate review of qualitative literature could complement the current study.”

21. Given the disability community's call to do "nothing about us without us," the team's limited plans to engage with disability advocacy groups, as well as with people with disabilities themselves in the study process and its dissemination seems like a potential missed opportunity or study limitation.

The study team includes a person with a disability, who has provided substantial input regarding study design.

We have added the following sentence regarding dissemination:

"Dissemination will also engage with disability advocacy groups, through the International Disability Alliance."

Reviewer 3:

1. The aims for the study are vast. Exploring the relationships between disability and violence against women is different than mapping definitions, measures and methodologies used to measure violence against women, in the context of disability-focused research. I am not sure which is the focus of the scoping review. Would clarify with what the research question is that you are looking to answer.

We recognize that the aims of this study are broad, which is appropriate for a scoping review. We have revised the Aims section to clarify the specific research question, which is mapping definitions, measures and methodologies in three interlinked bodies of literature (which in turn will help to explore relationships between disability and violence).

The Aims section now reads:

"This scoping review seeks to strengthen and support efforts to understand the linkages and intersections between disability and violence against women, specifically by mapping definitions, measures and methodologies in quantitative literature on this topic.

Violence against women is defined by the United Nations as "any act of gender-based violence that results in, or is likely to result in, physical, sexual or mental harm or suffering to women, including threats of such acts, coercion or arbitrary deprivation of liberty, whether occurring in public or in private life" ²². The World Health Organization defines disability as described in the International Classification of Functioning, Disability and Health [ICF]: disability is the outcome of the interaction between (1) one's health conditions and (2) contextual factors such as physical accessibility of the environment, access to assistive products, or attitudes of others. To describe a person's disability status under the ICF framework, it is necessary to examine both components of this definition ²³.

Our scoping review will map definitions, measures and methodologies in three areas of measurement of violence against women and disability: i) measurement of violence within the context of disability-focused research, ii) measurement in research focused on the intersection of disability and violence, and iii) measurement of disability in the context of research focused on violence against women. We focus on quantitative literature given our scoping review emerges from data requirements for the SDGs and seeks to address current developments in quantitative population-based surveys of violence against women. For the purpose of our review, we define disability-focused research as

quantitative research seeking to estimate the prevalence of disability or identify associations between disability and other health outcomes. We define research focused on the intersection of disability and violence as research that focuses on associations between disability and violence, without being solely focused on either disability or violence as an outcome. This focus on three distinct, but overlapping, areas of literature is designed to inform current debates and discussions regarding how to generate evidence concerning violence against women with disabilities. As noted above, for example, donors' interest in understanding the association between disability and violence against women has focused on incorporating measures of disability within national violence against women surveys. However, a broader characterization of which measures of disability and violence are used and available, how definitions are operationalized, and what methodologies are feasible and appropriate is needed."

2. Methods: Include who developed the search strategy/if a librarian or expert was consulted to do so.

We have added the following sentence:

"An expert librarian at the World Health Organization provided advice on search strategy and selection of databases."

3. What is the justification for 15 and older? A 15 year olds experience will likely be quite different than a 30 year olds experience.

We agree that a 15 year olds experience and a 30 year olds experience will be very different, and these differences can be analysed in the thematic synthesis stage.

We have included the following rationale:

"15 and above is selected as the age cut-off as this is lower age-limit for relevant SDG indicators."

4. For data management. Covidence or a similar program would be helpful to complete the screening process.

All members of the team do not have access to Covidence, so we are unable to use it as our screening program.

5. I suggest following the steps of Arksey and O'Malley for methods and outlining your methods with the use of their steps to conducting a scoping review, or another framework for conducting a scoping review.

We have drawn on Arksey and O'Malley's framework within the current structure of our Methods section, which we believe best fits a published protocol manuscript. For example, we have added references to their framework throughout, i.e. "This data extraction process corresponds with Stage 4, Charting the data, in Arksey and O'Malley's framework."

6. Include a section for study limitations.

Our limitations section is the final paragraph of the Discussion section:

"A primary limitation is the definition of disability that is operationalized in the review. The World Health Organization recognizes that disability includes "impairments, activity limitations, and participation restrictions," and that disability "is a complex phenomenon, reflecting the interaction between features of a person's body and features of the society in which he or she lives" (<https://www.who.int/topics/disabilities/en/>). According to this definition, knowledge of the environment in which a person lives is critical to understanding someone's experience of disability. However, to keep the scope of the review manageable and with the understanding that research on

the environmental component of disability is often lacking, we will utilize search terms for disability that primarily focus on impairments or specific health conditions that are known to cause particular impairments. This may limit the literature identified and included in the search and bias results towards research focused on only the bodily, health or impairment component of disability. A limitation of the process of study screening and selection is that one team member will conduct the grey literature search and identification process. This may limit the rigor of the grey literature search process, but will also enable review of more grey literature for possible inclusion (i.e. 10 pages of results for 3 separate searches). A limitation of the study design is the focus on quantitative literature. Qualitative descriptions are important components of understanding women’s experiences of disability and violence. However, the focus of this scoping review is specifically on quantitative measurement. This is motivated by the existing state of the field of evidence and need for data to answer key questions to inform prevention and response in violence against women policy and programs. A separate review of qualitative literature could complement the current study.”

VERSION 2 – REVIEW

REVIEWER	Jennifer MacGregor Western University, Canada
REVIEW RETURNED	19-Aug-2020

GENERAL COMMENTS	This is the second time I have reviewed this manuscript. The authors have addressed all of my initial concerns and I am satisfied with the other changes made as well. I recommend the article for publication.
---

REVIEWER	Rebecca J. Macy University of North Carolina at Chapel Hill, United States
REVIEW RETURNED	29-Aug-2020

GENERAL COMMENTS	I appreciate the authors' attention to the prior reviewer comments, which have significantly improved the manuscript. This protocol will make an important contribution to the literature concerning how rigorous, meaningful scoping reviews may be well conducted in the area of gender-based and interpersonal violence.
---